# Deep Learning-Based Simultaneous Temperature- and Curvature-Sensitive Scatterplot Recognition

**DOI:** 10.3390/s24134409

**Published:** 2024-07-07

**Authors:** Jianli Liu, Yuxin Ke, Dong Yang, Qiao Deng, Chuang Hei, Hu Han, Daicheng Peng, Fangqing Wen, Ankang Feng, Xueran Zhao

**Affiliations:** 1School of Mechanical Engineering, Yangtze University, Jingzhou 434023, China; 201972315@yangtzeu.edu.cn (J.L.); 2021720647@yangtzeu.edu.cn (X.Z.); 2School of Electronic Information and Electrical Engineering, Yangtze University, Jingzhou 434023, China; 2022720695@yangtzeu.edu.cn (Y.K.); heichuang@yangtzeu.edu.cn (C.H.); 201606279@yangtzeu.edu.cn (A.F.); 3School of Petroleum Engineering, Yangtze University, Wuhan 430100, China; hancupb@126.com; 4Key Laboratory of Exploration Technologies for Oil and Gas Resources, Yangtze University, Ministry of Education, Wuhan 430100, China; pengdc_geo@126.com; 5Hubei Key Laboratory of Intelligent Vision Based Monitoring for Hydroelectric Engineering, China Three Gorges University, Yichang 443002, China; wenfangqing@ctgu.edu.cn

**Keywords:** fiber optic sensor, scatterplot, finite element method, deep learning, temperature recognition

## Abstract

Since light propagation in a multimode fiber (MMF) exhibits visually random and complex scattering patterns due to external interference, this study numerically models temperature and curvature through the finite element method in order to understand the complex interactions between the inputs and outputs of an optical fiber under conditions of temperature and curvature interference. The systematic analysis of the fiber’s refractive index and bending loss characteristics determined its critical bending radius to be 15 mm. The temperature speckle atlas is plotted to reflect varying bending radii. An optimal end-to-end residual neural network model capable of automatically extracting highly similar scattering features is proposed and validated for the purpose of identifying temperature and curvature scattering maps of MMFs. The viability of the proposed scheme is tested through numerical simulations and experiments, the results of which demonstrate the effectiveness and robustness of the optimized network model.

## 1. Introduction

Fiber optic sensors operate by managing the optical properties of light signals during fiber optic transmission, such as the intensity, polarization state, and phase. Changes in these properties reflect and denote variations in environmental parameters like temperature, strain, and pressure [1]. Using coherent light signals to navigate through a multimode fiber (MMF), separate propagation modes emerge based on phase differences and propagation pathways, generating scattering patterns at the fiber’s end [2,3]. The effective refractive index under a propagation mode and any external influences affecting that refractive index determine the scattering pattern. Therefore, studying and measuring the scattering pattern is a feasible method of capturing the features corresponding to the distinct propagation modes and thus gaining insight into light transmission [4].

Light-scattering fiber optic sensors, a popular subclass of interferometric fiber optic sensors, can derive waveguide state information through multimode interference analysis. By overcoming the limits of system integration and demodulation speed [5,6], these sensors prove to be the most promising future direction in fiber optic sensing [7]. Applications of light-scattering fiber optic sensors include temperature [8,9], current [10,11,12], displacement [13,14,15], and vibration [16,17,18,19] detection, and the applications of fiber optic speckle sensors have advanced significantly. However, since MMFs are affected by factors including the external environment, length, and diameter [20], it is critical to accurately model the speckle pattern changes under conditions of perturbation. Despite speckle distribution differences with experimentally derived speckle maps, the simulated speckle maps show patterns that evolve with temperature, which is close to the actual phenomenon [21]. Since the small changes in the scattering pattern at the output are difficult to recognize with the naked eye, developing an efficient computational architecture for feature extraction and precise scatter pattern recognition has become a pressing necessity.

Currently, convolutional neural networks (CNNs) have transcended disciplinary boundaries and have become widely used in image classification and recognition. Extensive surveys and empirical studies have shown the efficiency, reliability, and superior performance of CNNs in curvature recognition from scatterplot data [21,22]. Considering the excellent performance of CNNs in tasks involving complex spatially structured data, this study selected this particular deep learning method for the task at hand. CNNs can be used to insightfully infer the correlation between inputs and scattering modes without prior sophisticated knowledge of fiber transmission properties and physics, thus more accurately explaining the mechanism of scattering mode transitions in MMFs [23,24]. However, few studies have been conducted on CNNs for temperature and curvature scattering map identification in optical fibers. To this end, this study numerically models fiber curvature and temperature perturbations based on the finite element method (FEM) and proposes a residual network with squeeze-and-excitation attention mechanisms (SE-ResNet) for identifying the temperature and curvature scattering maps of multimode optical fibers. This study helps in the feature extraction and correlation analysis of simulation-generated images in order to discern the link between scattergram brightness, darkness fluctuations, and the level of external interference.

## 2. Materials and Methods

### 2.1. Principle Analysis

Determining the scattering modes at the endpoint of an MMF under various temperatures and curvatures frequently necessitates solving specified vector fluctuation equations in order to help derive the development or propagation properties of the electric field $\vec{E}$ for each identified mode [25,26,27]:(1)$$\nabla \times \nabla \times \vec{E}-k_{0}^{2}n^{2}\vec{E}=0$$

The above equation includes the radial electric field $\vec{E}$, and the wave number in vacuum $k_{0}=2\pi /\lambda$. Furthermore, $n=n\left(x,y\right)$ denotes the radial refractive index distribution, which can be articulated as follows:(2)$$n=\left[\begin{array}{ccc} n_{x} & 0 & 0\\ 0 & n_{y} & 0\\ 0 & 0 & n_{z} \end{array}\right]$$
where parameters $n_{x}$, $n_{y}$, and $n_{z}$ depend on the combined effect of the thermo-mechanical perturbation $\Delta n_{i}^{TM}$, the thermo-optical perturbation $\Delta n_{i}^{TO}$, and the initial isotropic refractive index of the material $n_{0}$ [28]:(3)$$n_{i}=\Delta n_{i}^{TM}+\Delta n_{i}^{TO}+n_{0},\left(i=x,y,z\right)$$

Thermo-optic perturbation $\Delta n_{i}^{TO}$ transpires due to temperature modulation, signified by $\Delta n_{i}^{TO}=C_{TO}\left(T-T_{0}\right)$, where $C_{TO}$ represents the thermo-optic coefficient. The thermo-mechanical $\Delta n_{i}^{TM}$ of the material is also temperature-induced, denoted by $\Delta n_{i}^{TM}=-\frac{n_{0}^{3}}{2}\alpha k\left(T-T_{0}\right)$, where α signifies the linear thermal expansion coefficient, k represents the aggregate of strain optical coefficients $p_{11}$ and $p_{12}$, and $T_{0}$ demarcates the material’s original temperature [29]. Hence, we have the following equations:(4)$$\left\{\begin{array}{l} n_{x}=\left[C_{TO}-\frac{n_{0}^{3}}{2}\alpha \left(P_{11}+P_{12}\right)\right]\left(T-T_{0}\right)+n_{0}\\ n_{y}=\left[C_{TO}-\frac{n_{0}^{3}}{2}\alpha \left(P_{12}+P_{11}\right)\right]\left(T-T_{0}\right)+n_{0}\\ n_{z}=\left[C_{TO}-\frac{n_{0}^{3}}{2}\left(2\alpha P_{12}\right)\right]\left(T-T_{0}\right)+n_{0} \end{array}\right.$$

With marginal temperature variations, the effect of thermo-mechanical disruptions becomes relatively insignificant, permitting the omission of terms, including the strain optical coefficient. Consequently, Equation (5) can be simplified, thereby elucidating the nature of refractive index variation in nearly temperature-isotropic materials.
(5)$$n_{x}=n_{y}=n_{z}\approx C_{TO}\left(T-T_{0}\right)+n_{0}$$

Two methods are commonly employed to elucidate the properties of optical fibers with curvature: the conformal mapping method and the beam propagation method. Both may effectively model the effect of fiber bending on light transmission and determine the refractive index of an equivalent straight fiber through a correction formula. The relevant calculations, based on prior work [30,31], provide a crucial theoretical basis for understanding the optical behavior of optical fibers with curvature.
(6)n′=n1+x′R

The fiber is bent with a bending radius R, and $n^{\prime }$ is its equivalent refractive index after bending, with $x^{\prime }$ being the distance from the fiber’s center in the bending plane.

### 2.2. Fiber Specklegram Analysis

This study simulates the temperature and curvature of the optical fiber using finite element analysis. To facilitate in-depth numerical simulation and analysis, the bent fiber is first equated with a straight one using the beam propagation method and the conformal mapping method [31,32]. After discretizing the fiber cross-section into free triangular arithmetic units (Figure 1), COMSOL Multiphysics is used to calculate the electric field. The important parameters of the MMF are shown in Table 1, and numerical simulation of the MMF is carried out based on these parameters. Figure 2 depicts the main steps in the numerical simulation. The electric field distributions in straight fibers and fibers bent along the x- and y-axes are shown in Figure 3, along with the electric field modes of the temperature and curvature perturbations.

According to Figure 3, the electric field components inside the core of the straight fiber exhibit uniform and symmetrical distributions. With the progressive growth of the refractive index, the energy is increasingly concentrated in the center, achieving nearly no energy loss transmission. In contrast, according to the simulated fiber losses achieved with fiber bending radii of 5 mm, 10 mm, 15 mm, 20 mm, and 25 mm, the bending loss of the bent fibers diminishes as the bending radius rises, as shown in Figure 4. In particular, the bending loss is particularly significant under bending radii below 15 mm, where the electric field energy is largely concentrated in the axial area of the bending radius. Of note, the effect of increasing the bending radius further than 15 mm on the bending loss diminishes exceedingly, and its performance difference with that of the straight fiber is insignificant.

To experimentally investigate the bending loss characteristics of optical fibers with different bending radii of 5 mm, 10 mm, 15 mm, 20 mm, and 25 mm, the bending loss of the optical fibers is calculated accordingly, as shown in Figure 5. A comparative analysis of the optical fiber loss under different bending radii shows that, when the bending radius is between 5 mm and 10 mm, the optical fiber shows obvious bending loss at the bending position. However, the fiber loss shows no significant change as the bending radius is increased to 15 mm, 20 mm, and 25 mm, and the loss curve coincides with that of the straight fiber. A comparative analysis of the simulation results (Figure 4) and experimental results (Figure 5) indicates that the loss results under the simulated fiber bending radii are highly consistent with the trend of those under the experimental fiber bending radii, fully verifying the effectiveness and accuracy of our scheme. On this basis, the critical bending radius of the fiber is set to 15 mm. In the subsequent fiber temperature and curvature scattering map identification, cases with the bending radius below and above this critical value are fully considered to ensure the accuracy and reliability of the results.

## 3. Experimental Principle and Method

### 3.1. Data Preparation

Figure 6 displays the experimental platform used for the multiphysical field test on the optical fiber. A 20 mm long MMF with a 50 μm core diameter is used on the platform in Figure 6. By controlling the single-mode fiber (SMF), the light from the semiconductor laser (Agilent 8164A) is coupled and transmitted into the MMF under test conditions. The beam emitted from the MMF passes through the objective lens (GCO-213) onto a CCD camera with a 1920 × 1200 pixel resolution (MER-231-413M-L).

The following experimental procedures were followed to construct a large speckle atlas set of the temperature and curvature of the optical fiber. First, one end of the MMF was fixed to ensure its stable and constant position, while the opposite end was adjusted as required using a moveable scale. By carefully modifying the movement length x of the adjustable scale, the distance h between the two endpoints of the MMF can be adjusted, with h = L − x, where L represents the original length of the MMF. Meanwhile, the bending radius R of the MMF is modified, and consequently varies its optical properties.

To determine the bending radius R of the optical fiber, this study employed the approximate calculation approach show in [34], which could relatively precisely estimate the bending radius under a given experimental setup based on the physical characteristics of the optical fiber and the experimental circumstances. Through this approach, the effect of fiber bending on the temperature and curvature speckle maps was quantitatively explained, providing significant support for further data analysis and interpretation.
(7)$$R\approx 1/\sqrt{24x/L^{3}}$$

According to Equation (7), the length x of the scale can be altered to derive 5 alternative positions with bending radii ranging from 5 mm to 25 mm. Precise position control was essential in order to study the temperature and curvature properties of the optical fiber in different bending states. With the desired bending radii, an adjustable temperature box was used to precisely regulate the experimental environment. Specifically, a temperature gradient with a 0.2 °C step was designed, and the speckle patterns of the optical fiber were captured under each temperature and curvature over a wide temperature interval ranging from 0 °C to 120 °C. This experimental setup aimed to comprehensively reveal the optical response of optical fibers at different temperatures and bending radii, providing a rich experimental basis for subsequent data analysis and theoretical modeling.

### 3.2. Processing Method

Empirical evidence suggests that, with classical deep learning architectures, performance plateauing or degradation often occurs as the depth of the network increases. This degradation is primarily attributed to the vanishing and exploding gradient problem that compromises the model’s ability to converge and achieve optimal training. The ResNet design ingeniously circumvents this challenge by incorporating ‘skip connections’ or ‘shortcut connections’. These connections shrewdly bypass or ‘short-circuit’ multiple intermediate layers, thereby fundamentally allowing for the successful training of networks with significant depths [35].

The SE module presented by Hu et al. [36] innovatively enhanced the modeling of interdependencies among convolutional feature channels, thus augmenting the overall performance of CNNs beyond their conventional capabilities. The SE module efficaciously mitigates the constraints of classic CNNs through the squeeze-and-excitation steps. In the squeeze stage, the module efficiently reduces computational complexity through global average pooling, thus transforming the high-dimensional convolutional layer outputs into a condensed feature descriptor. The excitation stage generates a channel-specific weight vector using a fully connected layer paired with a non-linear activation function. These weights are deliberately allocated to every channel in the initial feature map, endowing each with a distinct importance score based on the features they communicate. Ultimately, the SE module permits the model to autonomously learn both the intrinsic weights and the relative relevance of multiple channels, adapting to enhance task-relevant variables while attenuating irrelevant ones. This dynamic feature priority modification enhances the discriminative power and leads to an automatic attention mechanism, culminating in improved model performance.

Figure 7 details the schema of the SE module and the foundational structure of the SE-augmented ResNet (SE-ResNet). By integrating SE modules into ResNet, the SE-ResNet structure provides an enhanced base for applications within this research scope. Further, SE-ResNet is enriched with the SE basic block to complement the foundational CNN framework. Figure 8 elucidates the architecture of SE-ResNet.

### 3.3. Performance Evaluation Metrics

Accurate model performance evaluation is crucial for recognition of the temperature and curvature of the speckle pattern dataset. Given the nature of this regression problem, the following four evaluation metrics are adopted to assess model performance. Each metric evaluates the discrepancy between the predicted and actual values from a different perspective, thereby providing unique insights. Combining these evaluation metrics, we comprehensively assess the regression model’s performance from multiple perspectives, including overall error, robustness, sensitivity to outliers, and worst-case performance. This holistic evaluation guides the improvement and optimization of the model. If there are m speckle images, $y_{i}=\left(y_{1},y_{2},\ldots ,y_{m}\right)$ is regarded as the true value score of the measured m images, and ${\hat{y}}_{i}=\left({\hat{y}}_{1},{\hat{y}}_{2},\ldots ,{\hat{y}}_{m}\right)$ is regarded as the corresponding estimation score. The evaluation indicators are defined as follows:

Mean Square Error (MSE) is defined as follows:(8)$$\mathrm{MSE}=\frac{1}{m}{\sum_{i=1}^{m}\left(y_{i}-{\hat{y}}_{i}\right)}^{2}$$

MSE measures the dispersion of the algorithm’s predictions from actual values by computing the mean of the squared deviations. The squared deviations minimize the biases of positive and negative errors, thus providing a broader indicator of general accuracy. As a renowned loss function in regression analysis, a smaller MSE generally signifies superior predictive accuracy and a desired algorithmic model. MSE quantifies the overall prediction error by averaging the squared errors. With the squared errors, MSE is more sensitive to outliers, effectively reflecting the precision of the predictive model in practical applications.

Mean Absolute Error (MAE) is defined as follows:(9)$$\mathrm{MAE}=\frac{1}{m}\sum_{i=1}^{m}\left|y_{i}-{\hat{y}}_{i}\right|$$

Taking the absolute value neutralizes differential errors’ positive and negative effects, yielding an intuitive appraisal of the prediction accuracy. Contrasting MSE, MAE exhibits higher sensitivity to extreme values, hence providing more context regarding the algorithm’s performance in diverse scenarios. MAE provides a simple and intuitive error measurement by averaging the absolute values of the errors. Since it does not involve squaring, MAE is less sensitive to outliers, making it an important metric for assessing the robustness of the model.

Root Mean Square Error (RMSE) is defined as follows:(10)$$\mathrm{RMSE}=\sqrt{\frac{1}{m}\sum_{i=1}^{m}{\left(y_{i}-{\hat{y}}_{i}\right)}^{2}}$$

RMSE enables a more intelligible interpretation than MSE, as its units are equivalent to the original data. A smaller RMSE indicates a lesser error in the model, i.e., superior accuracy. Employing MAE and RMSE concurrently offers insight into the sampling error spread. RMSE retains the sensitivity of MSE to large errors to some extent while providing a direct error measurement in the original units, making it suitable for use in evaluating overall performance.

The maximum error is defined as follows:(11)$$\mathrm{Maximum}\;\mathrm{Error}=\max\left|y_{i}-{\hat{y}}_{i}\right|$$

The maximum error refers to the greatest absolute difference between predicted and actual values. This metric emphasizes the worst-case scenario, thus shedding light on the algorithm’s predictive prowess under potentially unfavorable or extreme conditions. As such, it enables the formulation of risk-averse strategies while employing predictions. These measures collectively create an efficient performance evaluation framework for algorithmic models, underscoring their strengths and weaknesses and hinting at the pathway to potential improvements.

## 4. Results and Discussion

### 4.1. SE-ResNet Model Validation

This study initially simulates the fiber optic scatterplot sensor’s response to external temperature perturbations. When the perturbating temperature exceeds 120 °C, the scatterplot’s correlation coefficients show a non-linear response, which is in line with the findings in [8,25]. Consequently, 601 scatterplot images were produced from the temperature and curvature numerical simulations of the fiber scatterplot, with various fiber bending radii and temperatures ranging from 0 °C to 120 °C in a step of 0.2 °C. The dataset is divided into three parts: 60% for training, 20% for validation, and 20% for testing.

The learning rate is one of the critical hyperparameters influencing the effectiveness of model training. In the gradient descent algorithm, the learning rate controls the step size of gradient updates and adapts automatically throughout the learning process. In this study, we conducted comparative experiments on optimizers and learning rates to determine the optimal initial learning rate and batch size under controlled conditions. The interplay between learning rate and batch size requires experimental validation to ascertain the best combination. Our experimental results indicated that the Adam optimizer outperformed other optimizers across all tested learning rates (0.01, 0.001, and 0.0001). Therefore, this paper presents the performance of different optimizers at a learning rate of 0.0001. The loss curves of the training and validation sets in Figure 9 demonstrate that the Adam optimizer is the most suitable for this model. The most appropriate batch size and learning rate must be selected through balancing. A comparison indicated that the optimal hyperparameter combination has an initial learning rate of 0.0001 and a batch size of 32. Based on this combination, we employed SE-ResNet to recognize the curvature and temperature of speckle patterns.

The model training hyperparameters are shown in Table 2. The Adam optimizer was utilized to train SE-ResNet, starting with an initial learning rate of lr = 0.0001. A batch size of 32 was adopted for batch training, and a 0.5 dropout rate was used before the model’s output. Keras was selected as the deep learning framework. With 100 training epochs, the scatterplot input size was standardized to 128 × 128 for model testing and training. Model validation was performed after every iteration to ensure quality. Due to the training dataset’s moderate size, great effort was taken to prevent model overtraining, thus improving generalization and reducing overfitting. The converged final model was kept for later use. The same method was adopted for the data collection needed for experimental validation. To maximize the model’s generalization capacity, further data upgrades were also applied, including image flipping, panning, brightness and darkness modifications, and noise addition. To highlight the inherent disparities in speckle brightness and blackness, the image data were converted to grayscale. The matching temperature was then annotated on each scatterplot to guarantee predicted identification through later deep learning techniques.

The SE-ResNet is then utilized to extract the characteristics of scattering spots from the scattering image in order to develop a feature model with temperature and curvature perturbation mapping capabilities. The experimental results firmly suggest that SE-ResNet has outstanding prediction and recognition capacity, which sets the foundation for its prospective application in temperature and curvature perturbation response demodulation. Figure 10 displays the training process with the attention mechanism, and Figure 11 visualizes the prediction outcomes of the model.

Figure 10 and Figure 11 demonstrate that the developed model is remarkably accurate in forecasting fiber speckle maps. A careful inspection of the related loss profiles during iterative training indicates that the model has the intrinsic potential to swiftly converge to a loss-stabilized state, regardless of the network depth. The limited performance differences between the training and validation datasets indicates the model’s excellent generalization capacity, which improves its effectiveness and robustness.

### 4.2. Visualization of the Model

The properties of the two scatter plots, extracted using the residual network structure (SE-ResNet) for different temperatures and bending perturbations, are presented in Figure 12. The left ones are the original input images for the neural network, while the right ones are the neural network output, highlighting the relevant features of the scatter plots extracted through the second SE block.

A careful and systematic visualization of this dynamic machine learning model via a comprehensive layer-by-layer dissection reveals that information extraction efficiency does not uniformly span across every convolutional kernel. The model comprises various channels, each of which generates multiple feature maps. As each convolutional kernel is tailored to learn disparate features, one can observe the diversely extracted scatterplot features after the training protocol involving all model channels, as depicted in Figure 13.

### 4.3. Comparative Analysis of Model Predictions and Experimental Results

To validate the developed model under conditions of temperature and curvature perturbation, a comparative performance assessment was conducted with various representative deep learning regression models. These included widely recognized and extensively implemented models such as GoogleNet, AlexNet, VGG-16, and VGG-19. All networks used in the experiments were trained without pre-trained weights. To ensure a fair comparative evaluation, similar training parameter settings were adhered to across all deep learning models. The comparative results and the predictions are presented in Figure 14. The recognition performance of GoogleNet deteriorates significantly around the temperature range of 110 °C to 120 °C, resulting in substantial prediction deviations. AlexNet exhibits stronger generalization ability than GoogleNet due to its ReLU activation function and local response normalization techniques. Its predictions cover a wider range, reducing errors in temperature predictions at the edges. Both VGG-16 and VGG-19 utilize 3 × 3 convolutional kernels and ReLU activation functions across all convolutional layers, enabling them to capture abstract data features more effectively. This refinement of speckle pattern information leads to temperature predictions that are closer to the ground truth, particularly in overall temperature prediction.

To offer a more holistic model performance analysis, a multi-metric approach was adopted. The temperature prediction capabilities of the deep learning models were scrutinized based on MSE, MAE, RMSE, and maximum error. The numerical results from this expansive analysis are presented in Table 3. Compared to other CNN-based networks, our model constantly exhibits the lowest MSE, MAE, RMSE, and maximum error. Compared to GoogleNet, which utilizes Inception modules, SE-ResNet reduces MSE by 1.76, MAE by 0.78, and maximum error by 2.16. Compared with AlexNet, which also employs ReLU activation functions, SE-ResNet reduces MSE and MAE by 1.39 and 0.62, respectively. Despite VGG-16’s low MSE, its high maximum error indicates poorer predictive performance in certain extreme scenarios. Although VGG-19 increases the depth of the network by adding convolutional layers, its MSE, MAE, RMSE, and maximum error increase compared to VGG-16. Thus, simply increasing network depth does not always improve model performance. The experimental results demonstrate that SE-ResNet maintains a stable performance while increasing network depth. In contrast to the VGG series networks, which emphasize the stacking of local features, SE-ResNet adeptly utilizes global contextual information through the effective pooling and redistribution of feature channel weights. This strategic approach enhances the model’s practical generalization capabilities and outstanding performance.

Our discussion aims to evaluate the usefulness of numerical simulation, the effectiveness of the experimental methods, and the precision of result validation in the application of deep learning techniques to the identification of temperature and curvature perturbations in optical fibers. Two examples with bending radii of 10 mm and 20 mm were selected for analysis and validation during the experimental testing in order to further pursue these goals. The temperature and curvature identification results obtained under the two distinct bending radii are displayed in Figure 15 and Figure 16. For more thorough model performance assessments, many of the model evaluation criteria in Table 4 were included, thereby providing a quantitative foundation with which to compare the various approaches. Thus, a strong theoretical and experimental basis can be provided for the precise identification of fiber temperature and curvature perturbations through these thorough investigations and evaluations.

Figure 11, Figure 15 and Figure 16 indicate that, under temperature and curvature perturbations, the model prediction results gradually approach the experimental data as the bending radius grows. This observation is further supported by the evaluation metrics shown in Table 4.

## 5. Conclusions

Based on the fluctuating optics module of COMSOL, a temperature and curvature FEM model of an MMF with a core diameter of 50 μm is created. By numerically simulating the temperature and curvature coupling effect of the fiber, the impacts of numerous parameters on the fiber scattering map are examined. An optimal end-to-end residual neural network model is designed to identify fiber temperature and curvature perturbation scattering maps. The primary conclusions are as follows:(1)Although the refractive index of an unbent optical fiber varies under external temperature perturbation, the core electric field components show a uniform and symmetric distribution, with essentially no fiber loss.(2)The fiber loss of a bent optical fiber decreases with the increase in the fiber bending radius. The critical bending radius is determined to be 15 mm via numerical simulation and experimental testing.(3)Under temperature and curvature perturbations, the fiber scattering map shows greater complexity, and the predicted temperature and curvature scattering map of the fiber gradually approaches the experimental data with the increase in the bending radius. When the bending radius exceeds 15 mm, the model prediction error is minor due to the decreased fiber loss. When the bending radius is below 15 mm, the fiber loss increases, leading to a corresponding increase in the model prediction error. In addition, both simulation and experimental findings demonstrate the effectiveness and practicality of utilizing deep learning algorithms to detect the scattering map of optical fibers under temperature and curvature perturbations.

## Figures and Tables

**Figure 1 sensors-24-04409-f001:**
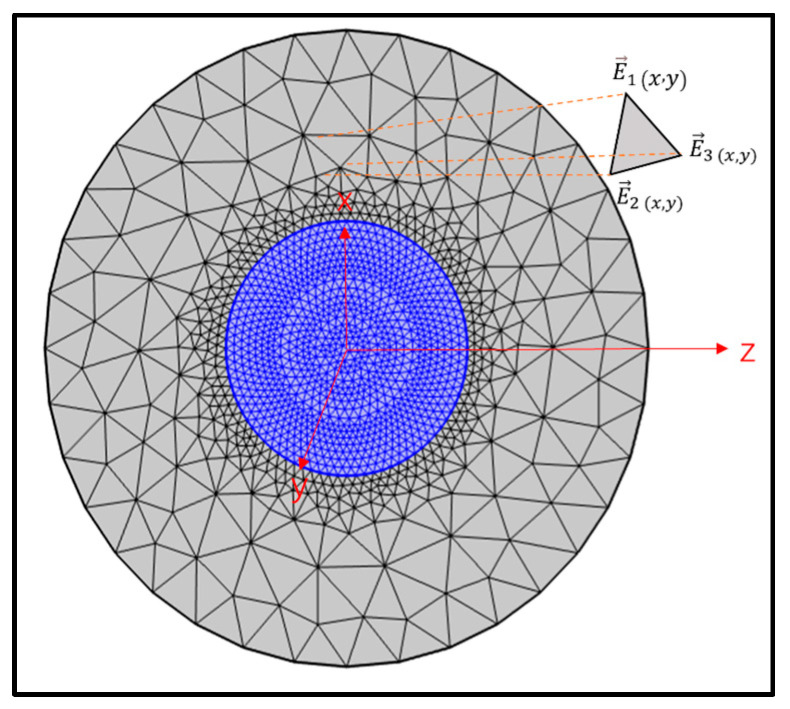
FEM cell mesh and node description of the fiber end surface.

**Figure 2 sensors-24-04409-f002:**
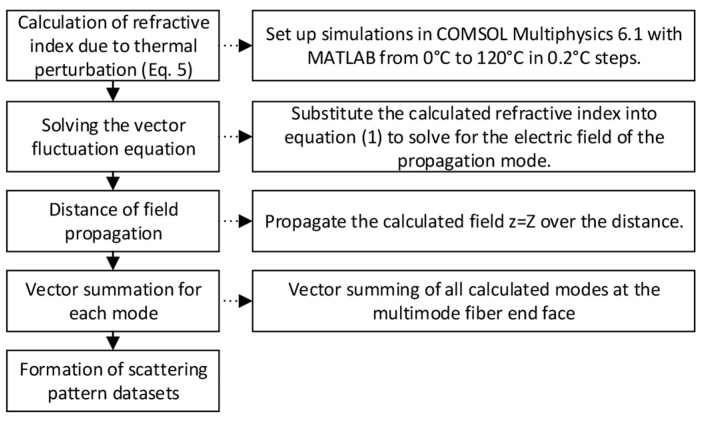
FEM-based scatterplot simulation flow chart.

**Figure 3 sensors-24-04409-f003:**
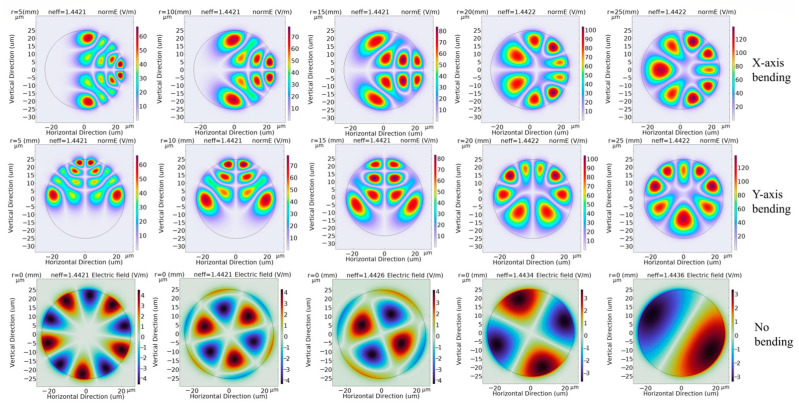
Electric field distribution and electric field modes in straight fibers and fibers with different bending radii and directions.

**Figure 4 sensors-24-04409-f004:**
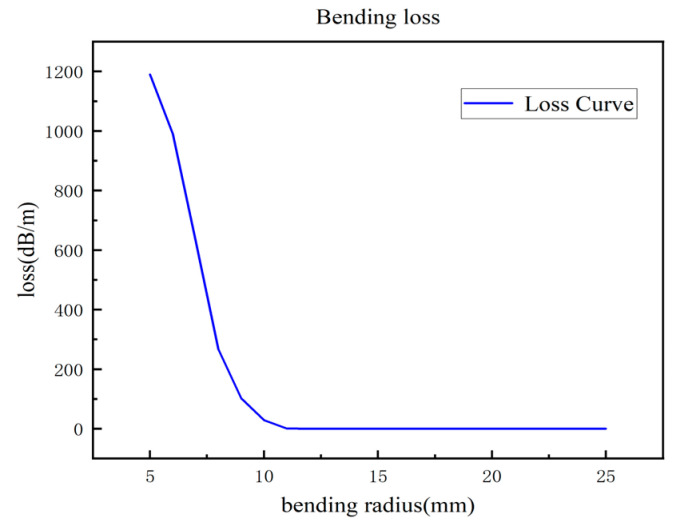
Numerical results of fiber bending loss at different bending radii.

**Figure 5 sensors-24-04409-f005:**
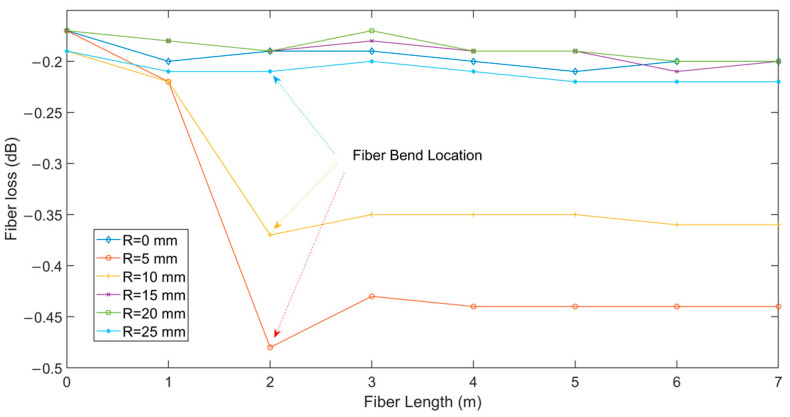
Experimental results of fiber bending loss at different bending radii.

**Figure 6 sensors-24-04409-f006:**
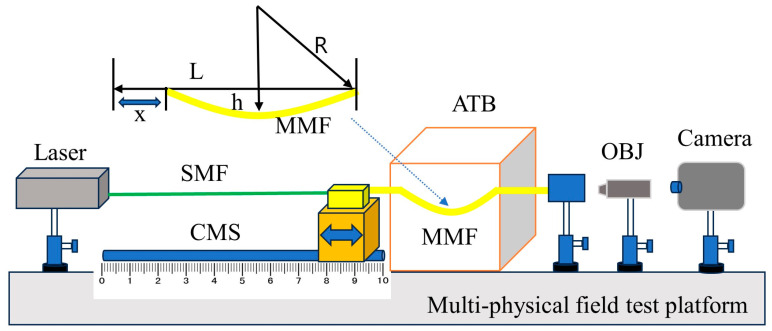
Schematic diagram of the temperature and curvature experiment platform. By altering the CMS’s (controlled moving scale) moving distance x, different bending radii R can be achieved. (OBJ: objective lens; ATB: adjustable temperature xox).

**Figure 7 sensors-24-04409-f007:**
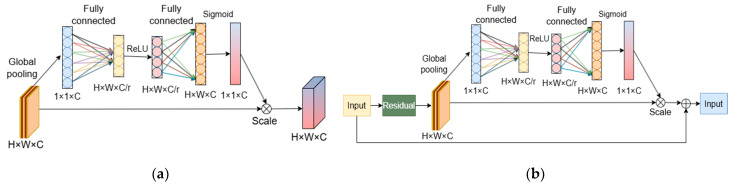
SE-ResNet combining residual block and SE module: (**a**) SE basic structure; (**b**) basic structure of SE-ResNet.

**Figure 8 sensors-24-04409-f008:**
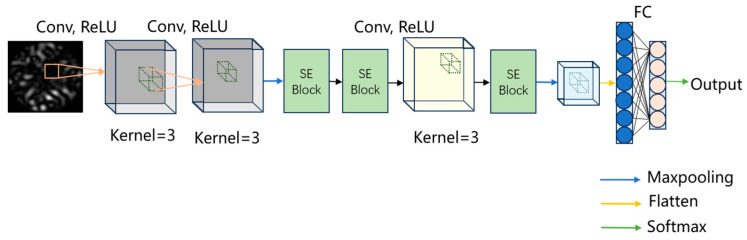
SE-ResNet model structure.

**Figure 9 sensors-24-04409-f009:**
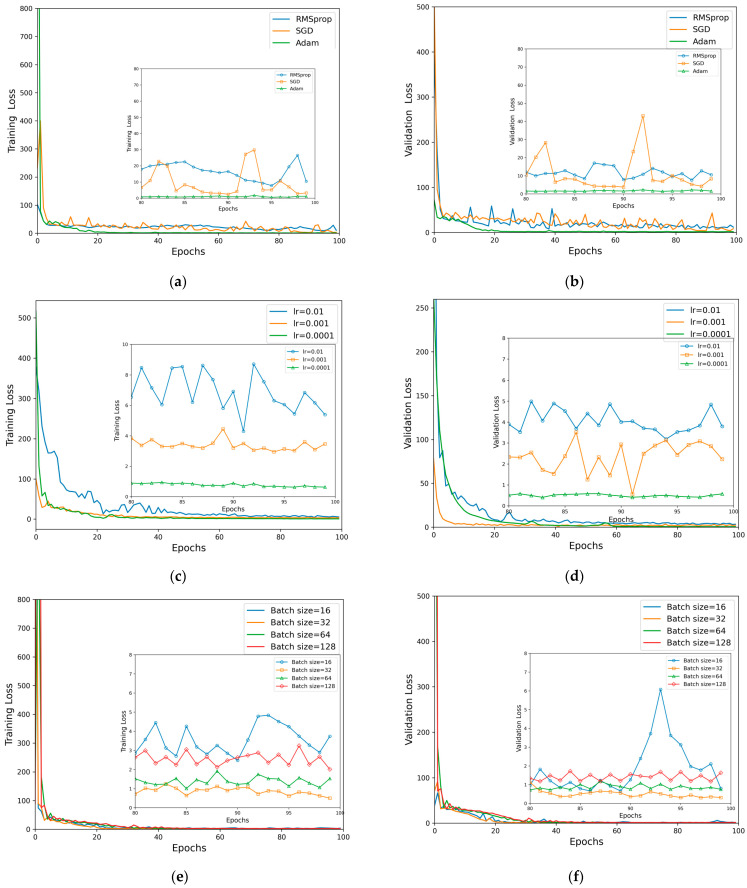
Training and validation curves using different algorithms and hyperparameters. (**a**) Training loss using different gradient optimization algorithms. (**b**) Validation loss using different gradient optimization algorithms. (**c**) Training loss using different initial learning rates. (**d**) Validation loss using different initial learning rates. (**e**) Training loss using different batch sizes. (**f**) Validation loss using different batch sizes.

**Figure 10 sensors-24-04409-f010:**
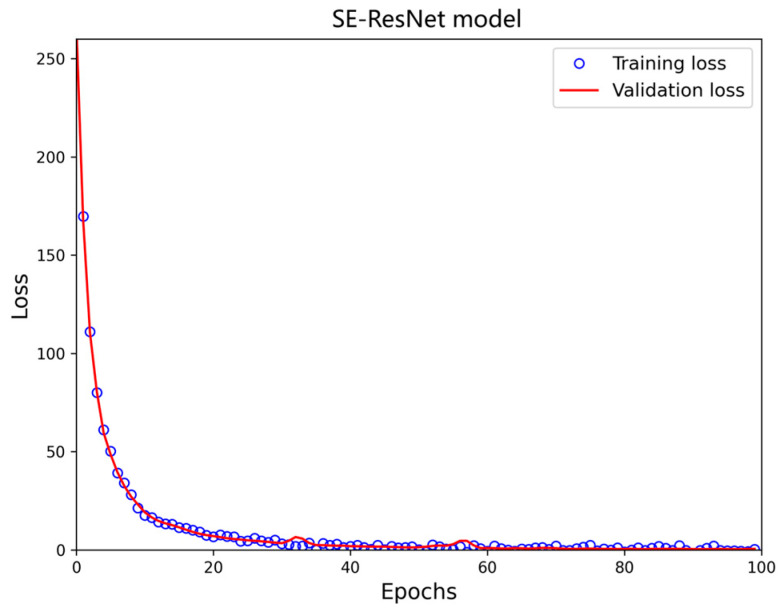
SE-ResNet training process.

**Figure 11 sensors-24-04409-f011:**
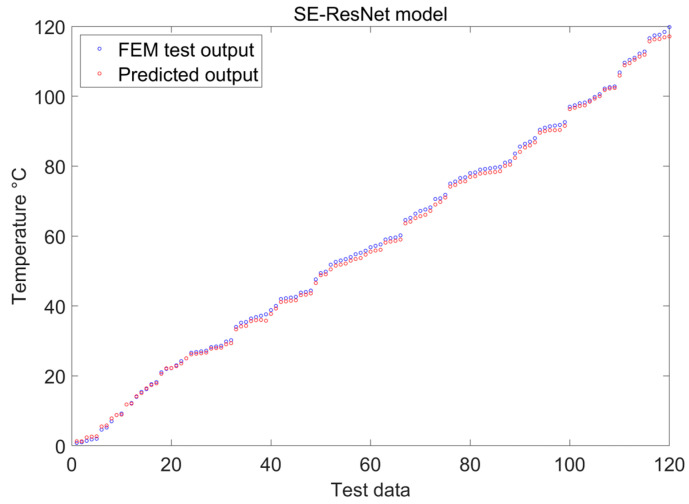
SE-ResNet prediction results.

**Figure 12 sensors-24-04409-f012:**
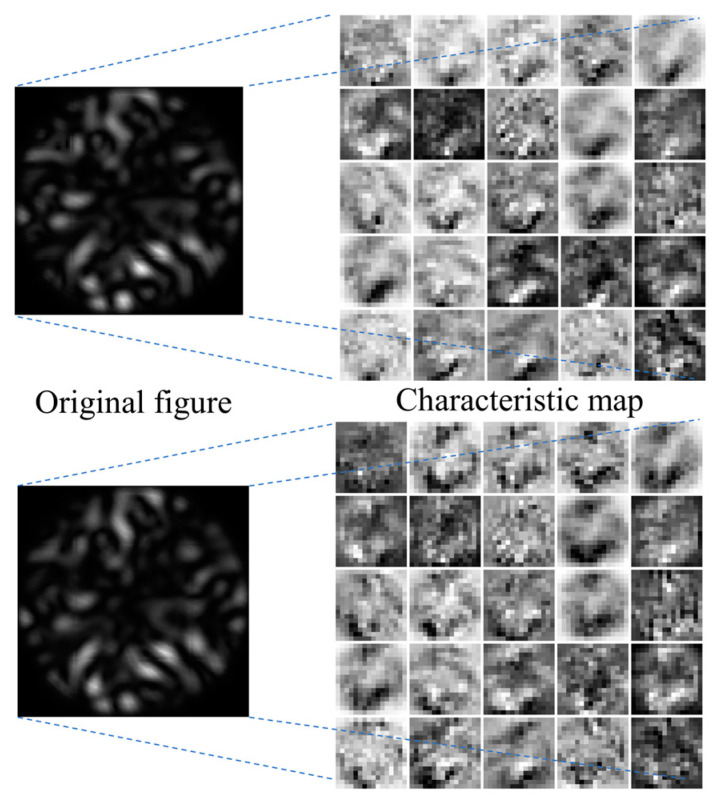
Visualization of SE-ResNet-extracted convolutional layer features.

**Figure 13 sensors-24-04409-f013:**
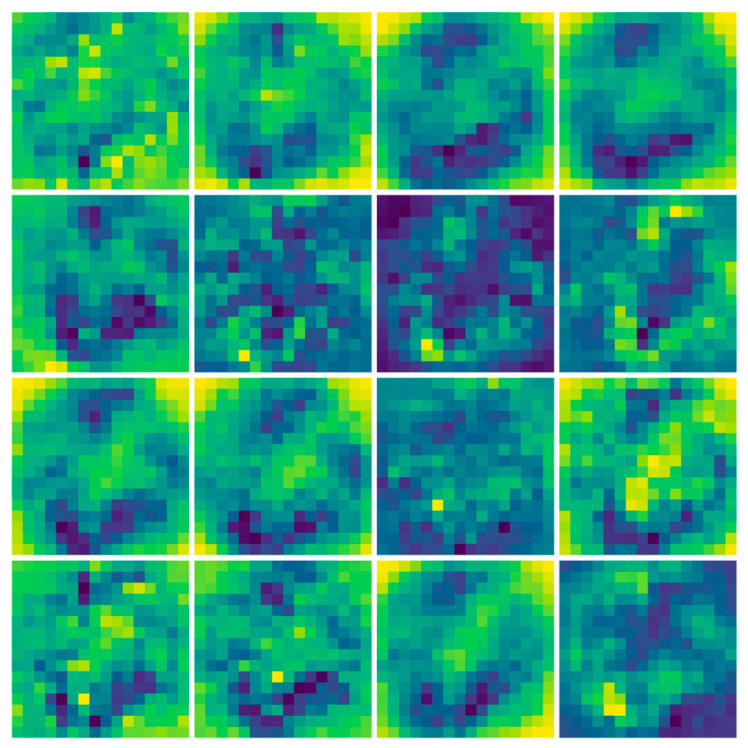
Visualization of SE-ResNet model features.

**Figure 14 sensors-24-04409-f014:**
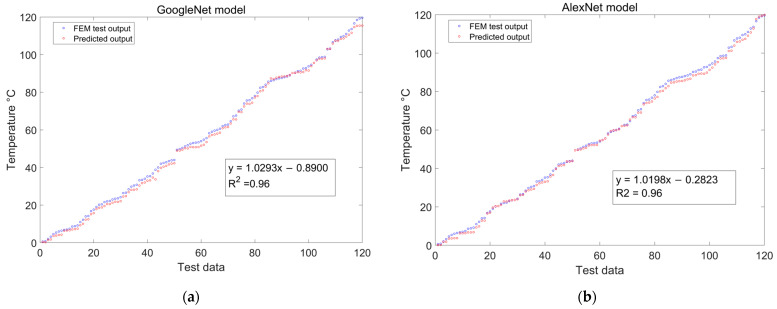
Prediction results of GoogleNet, AlexNet, VGG-16, and VGG-19: (**a**) GoogleNet; (**b**) AlexNet; (**c**) VGG-16; (**d**) VGG-19.

**Figure 15 sensors-24-04409-f015:**
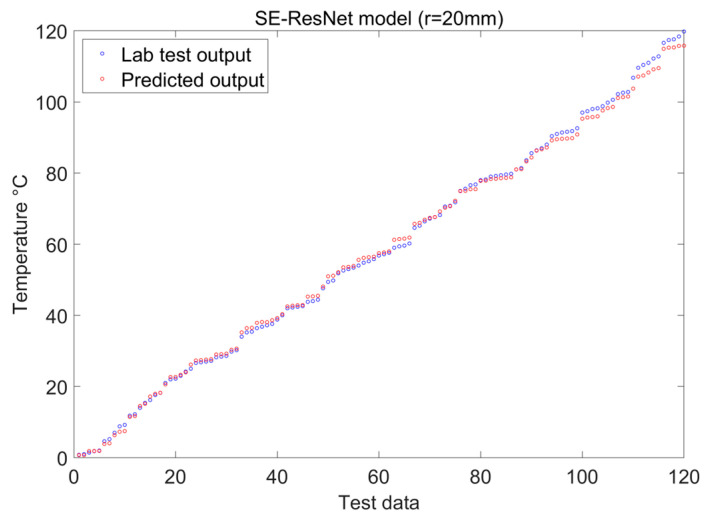
Visualization of SE-ResNet features with 20 mm bending radii.

**Figure 16 sensors-24-04409-f016:**
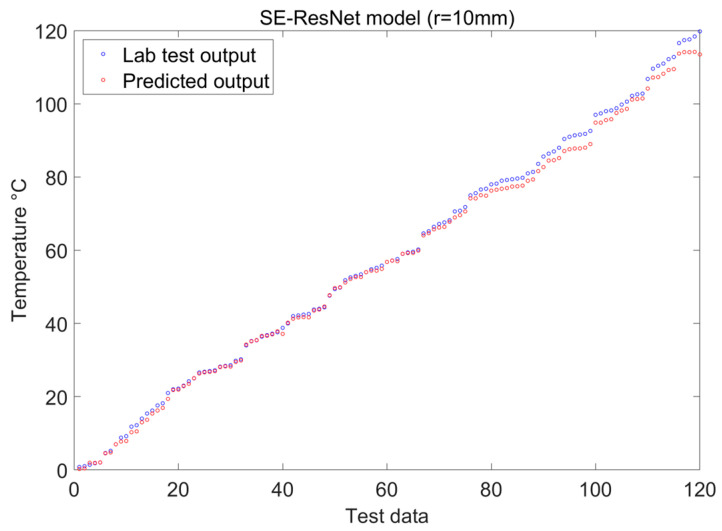
Visualization of SE-ResNet features with 10 mm bending radii.

**Table 1 sensors-24-04409-t001:** Simulation parameters of the specklegram dataset.

Parameters	Core	Cladding
Diameter	50 μm	125 μm
Refractive index	1.4437	1.4288
Thermo-optical coefficient	−10 × 10^−6^ *	10.5 × 10^−6^ *
Numerical aperture	0.22	
Wavelength	1490 nm	

* Parameters based on [33].

**Table 2 sensors-24-04409-t002:** Hyperparameters of the model.

Hyperparameter	Value
Activation function	ReLU
Optimizer	Adam
Learning rate	0.0001
Dropout	0.5
Batch size	32
Epoch	100

**Table 3 sensors-24-04409-t003:** Evaluation parameters of the model.

Model	MSE	MAE	RMSE	Maximum Error
SE-ResNet	0.21	0.37	0.45	1.15
GoogleNet	1.97	1.15	1.40	3.31
AlexNet	1.60	0.99	1.26	3.27
VGG-16	0.89	0.78	0.94	2.33
VGG-19	1.08	0.80	1.04	2.85

**Table 4 sensors-24-04409-t004:** Model evaluation parameters.

Model	MSE	MAE	RMSE	Maximum Error
SE-ResNet (FEM)	0.94	0.88	0.97	2.68
SE-ResNet (R = 25 mm)	1.18	0.82	1.14	2.87
SE-ResNet (R = 20 mm)	1.73	1.05	1.31	3.99
SE-ResNet (R = 15 mm)	2.97	1.51	1.72	5.11
SE-ResNet (R = 10 mm)	3.04	1.27	1.74	6.30
SE-ResNet (R = 5 mm)	7.17	2.04	2.68	8.62

## Data Availability

The data that support the findings of this study are available on request from the corresponding author. The data are not publicly available due to privacy or ethical restrictions.

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
