# Peer review of "Deep Learning-Based Simultaneous Temperature- and Curvature-Sensitive Scatterplot Recognition"

_sensors, 2024, doi:10.3390/s24134409_

Round 1

Reviewer 1 Report

Comments and Suggestions for Authors

The authors proposed the deep-learning based temperature and curvature recognition for a MMF. The detail investigations including the modeling, the simulation and the experiment were included. Based on the results, the validation on the proposed scheme has been confirmed. I suggest to accept the manuscript with the minor revision:

1. The reference for Table 1 was suggested to confirm the MMF data.

2. The simulation results for Fig. 5 were suggested to be included to confirm the validation of the simulation scheme.

3. How may data for training? How may data for testing? These information was suggested to be given in very clear place.

Reviewer 2 Report

Comments and Suggestions for Authors

The study simulates the impact of temperature and curvature on light transmission in multimode fibers (MMFs) using finite element methods, revealing complex interactions at input and output.  It identifies a critical bending radius of 15 mm for the fiber and creates speckle patterns for different temperature and curvature conditions.  An efficient residual neural network model is also introduced for automatically recognizing scattering features in MMFs, which is proven effective for detecting temperature and curvature changes through simulations and experiments. I will give some suggestions based on the manuscript.

1.      What are the characteristics of MMF with the change of set temperature in different bending states?

2.      How are the hyperparameters for model training Settings selected?

3.      The choice of performance indicators, including MSE, MAE, RMSE and maximum error, is notable. However, the manuscript does not provide the rationale for selecting these specific metrics. It would be beneficial for the readers if the authors could elucidate the reasoning behind choosing these particular indicators to evaluate model performance.

4.      The paper does not provide the data that the neural network model is designed to recognize, nor does it detail the recognition process. It would be beneficial for the reader to have a comprehensive understanding of these aspects.

5.      The proposed neural network model performs well in MMF recognition, and a graph comparison is also made. However, the article still does not explain how this method is superior to other identification methods. Highlighting the uniqueness of the work will enhance the appeal of the work.

6.      It seems that figures 7 to 12 in the article are a little vague, whether the author should increase their clarity.

Round 2

Reviewer 2 Report

Comments and Suggestions for Authors

This manuscript could be accepted.